# Ecosystem leaf area, gross primary production, and evapotranspiration responses to wildfire in the Columbia River Basin

Mingjie Shi[1], Nate McDowell[1,2], Huilin Huang[1], Faria Zahura[1], Lingcheng Li[1], Xingyuan Chen[1]

1. Pacific Northwest National Laboratory

2. School of Biological Sciences, Washington State University, PO Box 644236, Pullman, WA 99164-4236

*Correspondence to*: Mingjie Shi (mingjie.shi@pnnl.gov)

## Abstract

Wildfires impact vegetation mortality and productivity and are increasing in intensity, frequency, and spatial area in the western United States. The rates of vegetation recovery after fires play a major role in the reestablishment of biomass and ecosystem functioning (e.g., structure, resilience, and productivity), but such recovery rates are poorly understood. Here we use remotely sensed data products from the Moderate Resolution Imaging Spectroradiometer (MODIS) to quantify the resistance and resilience of leaf area index (LAI), gross primary production (GPP), and evapotranspiration (ET) to 138 wildfires with various burn severity across the Columbia River Basin of the Pacific Northwest in 2015. Increasing burn severity caused lower resistance and resilience for all three variables. Resistance and resilience are highest in grasslands, intermediate in savanna, and lowest in needleleaf evergreen forests, consistent with the adaptation of these vegetation types to fire. LAI has consistently lower resistance and resilience than GPP and ET, which is consistent with physical and physiological mechanisms that compensate for reduced LAI. Resilience is influenced by precipitation, vapor pressure deficit (VPD), and burn severity across all three vegetation types, however, burn severity plays a more minor role in grasslands. Increasing

wildfire severity will reduce the resistance and resilience and lengthen the recovery time of vegetation structure and fluxes with climate change, with significant consequences on the provision of ecosystem functioning and implications for model predictions.

## 1.    Introduction

While agricultural-activity changes reduce the global burned area (Andela et al., 2017), climate change increases the frequency and severity of wildfires (Burton et al., 2023; Jones et al., 2020; Pechony et al., 2010). The western United States is experiencing increasing numbers of fire events, with the growth in burned areas and number of severe fires over the past 40 years (Juang et al., 2022; Parks and Abatzoglou, 2020; Schoennagel et al., 2017; Westerling et al., 2016). Wildfires cause large impacts on ecosystem carbon and water cycles that last for decades (Adams et al., 2012; Bart et al., 2020). They change ecosystem structure and species composition, and modify soil properties, resource availability, the energy budget, and carbon storage (Anderegg et al., 2022; Bart et al., 2020; Lasslop et al., 2020; Turner, 2010). These changes manifest in vegetation-driven hydrologic changes in evaporation, transpiration, canopy interception, and indirectly alter infiltration, runoff, groundwater recharge, and streamflow (Adams et al., 2012; Partington et al., 2022). The ecohydrological impacts of wildfires are expected to increase due to the consistently observed (Williams et al., 2019) and predicted (Rammer et al., 2021; Wimberly et al., 2014) increase of wildfire frequency and severity under climate change.

A primary mechanism underlying the ecohydrologic changes from wildfire is the loss of ecosystem-scale leaf area i.e., leaf area index (LAI) (Shrestha et al., 2024), that reduces gross primary production (GPP) and evapotranspiration (ET) through the loss of photosynthetic and transpiring surface area and microclimate shifts (Collar et al., 2021; Li et al., 2018; Hu et al., 2008).

The scale changes in LAI, GPP, and ET cascade down to numerous consequences including reduced carbon storage and altered streamflow (McDowell et al., 2023; Seidl et al., 2014). Regional studies indicate that fire severity and post-fire plant community composition are essential to the spatial- and temporal-variations of ET (Collar et al., 2021; Poulos et al., 2021), where GPP and ET are tightly coupled in determining the ecohydrological processes (McDowell et al., 2023).

Resistance and resilience are the two metrics broadly used in ecology to evaluate recovery after disturbances. Resistance represents the ability of an ecosystem to withstand disturbance, while resilience represents the capacity of a system to recover its structure and function after a disturbance (Holling, 1973). Previous studies have used ecosystem traits, such as diameter at breast height, canopy height, live saplings (Proença et al., 2010), individual survival in fire, and pre- and post-fire bark thickness (Rodman et al., 2021) to estimate resistance and resilience due to fire-induced disturbance in forest plots. Wu and Liang (2020) explore both fire and drought determined resilience using two LAI products in different biomes at the global scale, suggesting the highest resilience in evergreen broadleaf forest. A continental United States based study implies that higher burn severities are consistently correlated with greater reductions in ET to precipitation ratio (ET/P), while the biggest magnitude of ET changes and percent of ET/P reductions often occur in evergreen forests and in shrublands (Collar et al., 2021). Using the Moderate-resolution Imaging Spectroradiometer (MODIS) GPP product, a Hurricane Rita based study suggests the difference in GPP recovery rates and resilience index between different vegetation types (Frazier et al., 2013). All these studies provide profound insights into disturbance impacts on ecosystem changes and regrowth (Albrich et al., 2020; DeSoto et al., 2020), implying that different vegetation types (VTs) may have different resistance and resilience determined by various disturbance intensity.

There have also been studies on the ecosystem-scale impacts and recovery of wildfires by using several different variables characterizing the ecosystem features (Jin et al., 2012; Mills et al., 2015). For example, a boreal forest based study indicates that aboveground biomass recovers slower than LAI and GPP, while GPP recovers quicker than LAI (Yu et al., 2023). Using different MODIS products, Marcos et al. (2023) show that vegetation and soil water content has higher

resistance but slower and more gradual recoveries than production. In general, the recovery processes can be regulated by post-fire environmental conditions. For example, using a predictive machine-learning model of time-to-recover, Rifai et al. (2024) reveal that leaf area recovery of forests is strongly accelerated by high post-fire precipitation anomaly. Currently, the immediate change and post-fire recovery of different ecosystem features (i.e., LAI, GPP, and ET) caused by

the same fire events, and their interactions with various burn severities (e.g., different burn severity categories; Eidenshink et al., 2007) have not been thoroughly quantified and compared, particularly at large scales (e.g., in river basins). Furthermore, the influence of environmental factors on post-fire ecosystem recovery, as characterized by LAI, GPP, and ET, remains inadequately quantified. This lack of clarity is closely linked to the uncertainties in predictions

made by Earth system models (ESMs; Lawrence et al., 2016). Thus, it is essential to evaluate the post-fire carbon and water fluxes, which are essential to the ecosystem recovery (i.e., returning to pre-fire carbon storage levels; Sun et al. 2020) and water resources (e.g., transpiration, streamflow; McDowell et al. 2023), respectively. An improved understanding of LAI, GPP, and ET responses to wildfire can benefit ESMs in term of reducing prediction uncertainties of wildfire impacts on

the carbon and water cycles.

This study aims to comprehensively investigate the post-fire ecosystem features (e.g., canopy carbon status versus the carbon and water fluxes) characterized by resistance and resilience

in the featured VTs of the Columbia River Basin (CRB), across 138 fires with different burn severities in 2015. Previous studies document the findings regarding the different recovery features

between VTs (DeSoto et al., 2020), the varied recovery features characterized by different variables (e.g., LAI, GPP, and ET) (Marcos et al. 2023; Yu et al. 2023), and the different responses of GPP to various environmental factors across VTs (Lu and Yan, 2023). Based on the existing research, we plan to test four research hypotheses: (1) higher burn severity results in lower resistance and resilience represented by all the three metrics across all the VTs, (2) with the same

burn severity, resistance and resilience are highest in grasslands, intermediate in savanna, and lowest in forests, (3) across all VTs resistance and resilience post-disturbance are highest for GPP and ET, and lowest for LAI, a major determinant of GPP (Saigusa et al., 2005), and (4) precipitation and VPD are more important to the resilience in grassland than in other VTs.

To test these hypotheses, we use the MODIS LAI, GPP, and ET products to quantify fire

induced changes in resistance and resilience, and the random forest feature importance method (Breiman, 2021) is used to investigate climate (e.g., precipitation, VPD) dependency. Here, we simultaneously and quantitatively examine the responses of LAI, GPP, and ET to fire disturbances by applying the resistance and resilience quantification framework discussed in DeSoto et al. (2020) to the MODIS products, in relation to burn severity and VTs (e.g., forests, savanna, and grasslands)

across the CRB in the Pacific Northwest, USA. Given that the impacts of wildfires are expected to intensify in CRB (Halofsky et al., 2020; Wimberly et al., 2014) and globally (Andela et al., 2017; Bowman et al., 2020; Jones et al., 2024), quantifying fire impacts is essential for both ecosystems and society. The research framework of this study can be broadly applied to quantify wildfire-induced ecosystem responses and evaluate the impacts of wildfires as revealed by

different data products and represented by ESMs.

## 2.    Methods

The analyses are performed at spatial resolutions 500–1000 meters, and our research time frame is 2011–2020, which is centered around the time of maximum fire occurrence of CRB in 2015. We use the (1) MODIS land cover type (LCT; Sulla-Menashe et al., 2018) to identify surface VTs; (2) burn severity product from the Monitoring Trends in Burn Severity (MTBS) program to classify the location and severity of fires (Eidenshink et al., 2007); (3) the meteorological data from ECMWF Reanalysis Version 5 (ERA5) to quantify annual variation in climate (Hersbach et al., 2020); (4) the MODIS LAI (Myneni et al., 2002), GPP, and ET products (Running et al., 2004) to assess the ecosystem resistance and resilience due to fire disturbance. To interpret the essential factors controlling resilience of different VTs, the random forest feature importance method is used to assess the importance of precipitation, VPD, and burn severity to the resilience values in 2020 represented by LAI, GPP, and ET. All the data and scripts for data processing are publicly available at the Environmental System Science Data Infrastructure for a Virtual Ecosystem (ESS-DIVE) repository (Shi et al., 2025).

### 2.1 Characterizing surface vegetation types

In this research, we use the annual 500 m MODIS LCT dataset, MODIS MCD12Q1 version 6.1 (Sulla-Menashe et al., 2018; Friedl et al., 2022), to identify the surface VT (Table S1). The VT map in 2015 shows that needleleaf evergreen forest (NEF), woody savannas (WDS), and grassland (GL), and croplands (CL) are the four dominant vegetation cover types over the CRB (Figure S1), and we study the impacts of wildfire over the NEF, WDS, and GL VTs.

## 2.2 Identifying the 2015 fire events

We identify all the 2015 fire events in the CRB so that we would have sufficient data during both pre- and post-fire for calculating resistance and resilience, and because 2015 is an extreme fire year in this region. The MTBS (1984–present) maps burn extent and severity across the United States (Eidenshink et al., 2007; Picotte et al., 2020). MTBS includes all fires $>= 4.05$ km$^2$ in the western United States, where burn severity is quantified as visible alteration of vegetation, dead

biomass, and soil that occurs within a fire perimeter (Eidenshink et al., 2007). Changes in vegetation status and biomass resulting from fires were assessed using the Composite Burn Index (CBI). These changes are also correlated with remotely sensed estimates such as the differenced Normalized Burn Ratio (dNBR), a metric measuring the difference between pre- and post-fire NBR images (Eidenshink et al., 2007). The burn severity product from MTBS is widely used as a

viable estimate of burn severity within certain ecosystems in the United States (Cansler and McKenzie, 2012; Picotte et al., 2020).

         The MTBS products include burn perimeters and burn severity, and we use the burn severity categories to identify fires and their features (e.g., burned area, burn severity) over the CRB. The MTBS mapping primarily relies on the differenced normalized burn ratio (dNBR)

algorithm and LandSat imagery in the near-infrared and shortwave infrared bands (Eidenshink et al., 2007). The same dNBR algorithm is applied to different VTs. Differenced NBR images— where post-fire NBR is subtracted from pre-fire NBR—are known as dNBR images. These dNBR images illustrate fire-related changes, which are categorized into severity classes, providing an unbiased foundation for analyzing additional fire effects (Eidenshink et al., 2007). The MTBS data

are at a 30-meter spatial resolution and upscaled to the 500-meter spatial resolution for the comparison with the MODIS data products (Table S1). MTBS employs different integers to

indicate burn severity categories, with values ranging from 1 to 4 representing unburned, low, moderate, and high severity, respectively. Consequently, the upscaling processes using the area-average remapping method produce floating-point numbers. Here, the numbers and meanings of burn severity values before and after the re-group are in Table S2. Based on this re-group method, the fire events and their burn severity in CRB are shown in Figures S2. To identify the vegetation type where each fire event occurred, we apply the MTBS fire boundary (i.e., shape) files, which describe the perimeter of each fire event, to the VT map (Figure S1). We use the dominant VT of each fire event, defined as the VT whose area accounts for more than 50% of burned area for that event, to identify the representative landscape type of each fire event (Figure S2b). This analysis aims to comprehend which VT(s) are predominantly affected by fire events across the CRB (Figure S2b). However, to precisely estimate the resistance and resilience of different VTs, we explicitly consider the VTs and their changes characterized by LAI, GPP, and ET in each data pixel of each fire event (see more details in Section 2.4). During the fire season of 2015, 138 fire events are identified. In all these burned areas, we remove the areas that experienced fire in 2011–2014 or in 2016–2020; thus, our resistance and resilience calculations are not confounded by repeat fires.

## 2.3 Interannual climate

We quantify interannual climate throughout the study region to determine if our resistance and resilience estimates were influenced by climatic variation. Here, we use precipitation, surface air temperature, and vapor pressure deficit (VPD) from ERA-5 (2011–2020; Hersbach et al. 2020). The dataset is originally at the 30-kilometer spatial resolution, and we use the nearest-neighbor method to downscale the data to the 500-meter spatial resolution to match the spatial resolutions of other datasets (e.g., MODIS) of this study. The ten-year mean precipitation and surface air

temperature are shown in Figure S3. We then use the MODIS LCT suggested VT and MTBS burn

severity information in each 500-meter data pixel to group precipitation, surface air temperature

and VPD within each fire disturbed region to their respective VT and then averaged the grouped

climate variables for each VT. The specific process is the same to the MODIS LAI, GPP, and ET

grouping, and more details of this method are introduced in the data description of MODIS data

products of LAI, GPP, and ET.

**2.4 Quantifying LAI, GPP, and ET**

We use the MODIS LAI product at the 4-day interval and 500-meter spatial resolution

(Myneni et al. 2002), and the MODIS GPP and ET products at the 8-day interval and 1000-meter

spatial resolution (Running et al., 2004), which was downscaled to the 500-meter spatial resolution

by using the nearest-neighbor method. To identify LAI, GPP, and ET changes among different

VTs and burn severity categories, we apply the MTBS boundaries and MODIS LCT suggested

VTs to the MODIS LAI, GPP, and ET products. To ensure the calculation accuracy, we evaluate

the variations of these metrics by using MODIS VT pixels within the fire boundaries to group

these variables based on VTs and calculated the means for the same VTs across all the fire

boundaries. Specifically, within each MTBS fire boundary, the MODIS VT information for each

data pixel is used to derive different VT determined LAI, GPP, and ET changes in the

corresponding MODIS data pixels (Table S1). We then average LAI, GPP, ET of the same VT and

with the same burn severity across all the 500-meter MODIS data pixels. As discussed above,

ERA5 precipitation and temperature data are also grouped between different VTs by using this

method. Thus, instead of considering the dominant VT in each fire boundary, we accurately

perform the calculation, which could avoid the errors associated with the weights of each VT in different fire boundaries. All the above-mentioned calculation are performed during 2011–2020.

**2.5 LAI, GPP, and ET based resistance and resilience calculations**

Following De Soto et al. (2020), we define resistance and resilience as:

$$\text{Resistance} = \frac{A_{2016}}{\bar{A}_{2011-2014}} \qquad (1)$$

$$\text{Resilience} = \frac{\bar{A}_{2017-2020}}{\bar{A}_{2011-2014}} \qquad (2)$$

where A represents the ecohydrological variables, LAI, GPP, and ET, used in this study, and the specific years are indicated in Equations (1) and (2). We exclude 2015 values of LAI, GPP, or ET in the calculations because the fires happened mid-way through the growing season (Figure S4), thus the 2015 values include both pre- and post-fire, making them inappropriate for resistance and resilience calculations. Given that resilience could exhibit interannual variations due to climate variations (e.g., DeSoto et al., 2020), we also calculate resilience for each individual year for all the VTs with various burn severity levels. DeSoto et al. (2020) define recovery as the condition following a disturbance compared to the condition in the year when the disturbance occurred, which can be represented by the following equation:

$$\text{Recovery} = \frac{\bar{A}_{2017-2020}}{A_{2016}} \qquad (3)$$

Thus, the primary distinction between resilience and recovery lies in the reference conditions: pre-fire versus the disturbed conditions. Resilience is defined as the capacity of a system to recover its

structure and function after a disturbance (Holling, 1973). To ensure clarity in our discussions of recovery status, we use the term resilience throughout this study.

We use LAI, GPP, and ET observations from the growing season, which we defined as days with values larger than 30% of the annual maximum. This threshold number can be tweaked (Shi et al., 2020), and we choose to use this value to avoid the MODIS data uncertainty during

snow seasons and minimize data noises. To avoid any error associated with using only a single observation, we identify the annual peak value and then averaged that value with records from the previous and subsequent eight days to generate the annual maximum value. This means that for MODIS LAI, with the 4-day temporal resolution, we average five contiguous records centered around the peak value. For MODIS GPP and ET, with the 8-day temporal resolution, we average

three records, one before the peak, the peak itself, and one after the peak. To obtain the start and end of the growing seasons, we calculate the 4-record running mean (i.e., 16 days) of LAI and 2-record running mean (i.e., 16 days) of GPP and ET over the entire year. The start of each year's growing season is determined when the running mean exceeds 30% of the annual maximum value, and the end of growing season was calculated with the running mean dropped below 30% of the

annual maximum. The growing season length based on different vegetation types with varied burn severity is shown in Figure S5.

### 2.6 Random forest feature importance

To interpret the factors controlling resilience of different VTs, the random forest feature

importance method (Breiman, 2021) is implemented using the scikit learn package in Python. Random forest uses a large collection of decision trees to predict the target variable based on its relationship with a specified set of input features. Each tree learns from a randomly chosen subset

of samples and features, while the final prediction is made by averaging predictions from all trees. Furthermore, the algorithm reports the relative importance of input features by considering the reduction in impurity achieved by each feature during tree construction.

For this analysis, the random forest is trained with a set of input features that include burn severity in 2015, and total precipitation and mean VPD between 2017 and 2020, for each grid in the burn areas. Nine separate models are trained to predict three target variables: resilience for LAI, GPP and ET in year 2020 for NEF, WDS and GL. The number of samples in NEF, WDS and GL were 11,881, 15,684 and 26,840, respectively.

Random forest hyperparameters such as number of trees and number of features considered at splitting are predefined before model training. Here, number of trees was set to 100. The GridSearchCV algorithm from the scikit learn package is applied on 85% of randomly chosen samples to find the optimal number of features considered at splitting, and it is determined to be 1. Model training and testing are performed by splitting the samples randomly with 85% in training and 15% in testing. The random forest model is trained 100 times by performing 100 randomized splits to reduce any bias from splitting, where the experimental design with 100 times training was well tested by previous studies (Juarez-Martinez et al., 2018; Sadler et al., 2018). From the 100 trained models, the distribution on train and test $R^2$ scores are obtained and the relative importance for each feature were averaged.

## 3.    Results

### 3.1 The meteorological conditions and burn severity in CRB

The MTBS and VT based analysis shows that August is the month with the highest fire frequency in 2015 with 91 fire events (Figure S2a), where NEF experiences 42, WDS experiences

27, and grassland experiences 67 fire events, respectively (Figures S1 and S2b). Northern CRB experiences a higher incidence of forest fires, whereas southern CRB is mainly affected by grassland fires (Figures S1, S2c, and S2c). There are two fire events in croplands, which were excluded from further analysis.

Mean precipitation and surface air temperature over the Columbia River Basin are 789 ± 63 (mm year$^{-1}$) and 5.7 ± 0.7 (°C) during 2011–2020 (Figure S3). The spatial pattern of precipitation and surface air temperature suggest a relatively warmer and drier condition in the southern part of the basin, where the areas are mostly covered by grassland (Figure S1). The western and northeastern areas of the basin have higher precipitation, ranging from 700 to 1300

mm year$^{-1}$, and lower air temperatures, ranging from -3.0 to 11.0 (°C) (i.e., from the northernmost part to the central–southern part of CRB; Figure S3b). These areas have a greater proportion of NEF and WDS (Figures S1 and S3). We further examine the climate for each of the 138 fire locations broken into the three vegetation types. Climate conditions in 2015, the year of high fire activity, is particularly dry and warm across all vegetation types. There is no significant difference

in mean annual precipitation and surface air temperature between 2011–2014 and 2016–2020 (Figure S4 and Table S3). Therefore, climate variations are not confounding resistance and resilience calculations.

### 3.2 LAI, GPP, and ET 2011–2020

Wildfires reduce LAI, GPP, and ET below the pre-fire mean in all VTs at the highest burn severity (herein $S_{burn}$; $S_{burn}$ >3; Figure 1; we present results for $S_{burn}$ below 3 in Figure S7 and Table S2). The 2011–2014 growing season mean LAI values are 1.87 ± 0.10, 1.47 ± 0.04, and 1.16 ± 0.03 m$^2$ m$^{-2}$ over NEF, WDS, and GL, respectively. The growing season LAI has an increasing

trend from 2016 to 2020 in all the VTs, with 2020 values of 1.18, 1.04, and 0.88 $m^2$ $m^{-2}$ for NEF,

WDS, and GL, respectively (Figure 1a). GPP and ET patterns are similar to those of LAI, with the

highest values during 2011–2014 and the lowest values in 2016. GPP and ET in 2020 are not back

up to the mean 2011–2014 values (Figures 1b and 1c). Similar but less dramatic declines in LAI,

GPP, and ET are observed in the lower burn severity classes (Table S2 and Figure S7). We also

calculate the standard error across the burned pixels of each VT by using all three metrics, i.e.,

LAI, GPP, and ET (Figures 1 and S7). The ratio between the standard error and spatial mean are

highest when $S_{burn} > 3$, and this ratio decreases with burn severity. This ratio also tends to decrease

with the ecosystem complexity with the highest values in NEF and lowest values in GL (Figure

not shown), and its ranges across the three VTs are 0.067–0.072 for LAI, 0.090–0.100 for GPP,

and 0.067–0.80 for ET, respectively, where the lower bounds are consistently representing the

values for GL. These results indicate a relatively high spatial variations of all the three metrics

when burn severity increases, and in the same burn severity category the spatial variations in LAI

are smaller than those in GPP and ET (Figures 1 and S7).

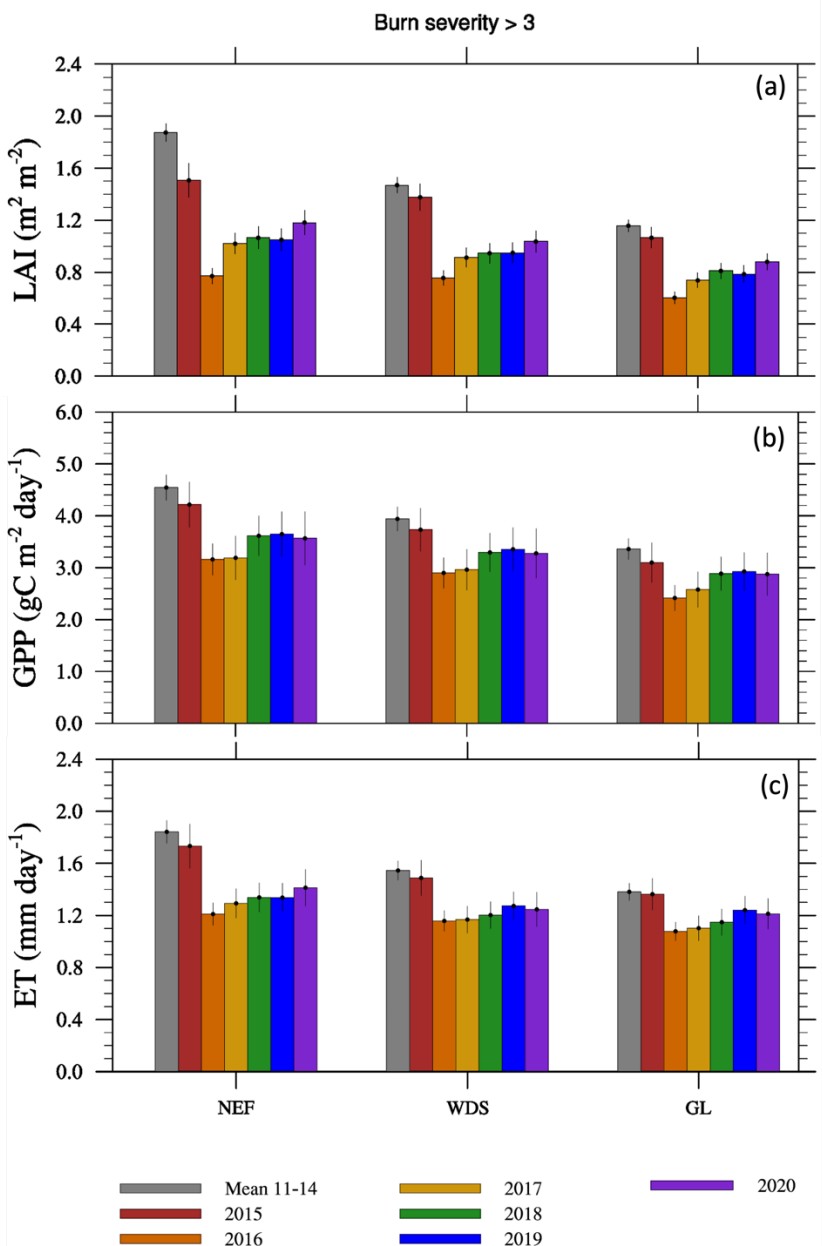

**Figure 1.** The growing season (a) LAI, (b) GPP, and (c) ET variation over needleleaf evergreen forests (NEF), woody savannas (WDS), and grasslands (GL) with burn severity >3 during 2011–2020. See Figure S7 for these results for fires with burn severities less than 3. The standard errors are included on the bars to represent the data variability across data pixels of different VTs.

**3.3 LAI, GPP, and ET resistance and resilience to wildfire**

We first present wildfire resistance and resilience for each VT (using equations (1) and (2)) across the burn severity categories and present the results as a function of time further below. Resistance to wildfire declines with increasing burn severity values for LAI, GPP, and ET, and is highest for GL, intermediate for WDS, and lowest for NEF VTs, regardless of response parameter

(i.e., LAI, GPP, or ET; Figures 2a, 2c, and 2e). Resilience to wildfire, calculated as the average resilience value from 2017–2020, is lower with higher burn severity for LAI, GPP, and ET (Figures 2b, 2d, and 2f). Like the patterns of resistance values, resilience is highest for GL, intermediate for WDS, and lowest for NEF. GL resilience has near 1 for all three variables in GL when burn severities are below 2.

Resistance and resilience calculated at the annual scale using equations (1) and (2) show the responses of LAI, GPP, and ET relative to each other (Figure 3; $S_{burn}>3$ shown, $S_{burn}$ values below 3 are shown in Figures S8–10). Within each VT, resistance and resilience are similar for GPP and ET, and are lower for LAI. Resistance and resilience increase for all parameters with lower burn severities (Figures S8–10), and are lowest for NEF, intermediate for WDS, and highest

for GL VTs. The standard errors obtained from the resistance and resilience values across different pixels indicate that LAI suggested resistance and resilience have relatively smaller spatial variations than those represented by GPP and ET. Given that resilience is closely related to water availability and burn severity (Figure 4 and the description below), the resilience under the same burn severity shows similar interannual variability (Figure 3). Furthermore, the recovery capability

(i.e., resilience) is similar for NEF and WDS when they are disturbed by the same burn severity (Figures 3a and 3b).

To examine the drivers of the interannual variation of resilience characterized by LAI, GPP, and ET, we use the random forest feature importance method to identify the contributions of

precipitation, VPD, and burn severity to influencing ecosystem resilience. Burn severity is more

important for NEF and WDS VTs than for GLs (Figure 4). In NEF, precipitation and VPD have importance scores of 0.3 for LAI resilience, while that of burn severity is 0.4 (Figure 4a). Similarly, in WDS, the importance scores of precipitation and VPD are 0.28 and 0.29, while that of burn severity is 0.43 (Figure 4b). Precipitation and VPD have relatively similar importance scores within VTs but were higher for GLs. In GL, the scores of precipitation, VPD, and burn severity to

LAI represented resilience are 0.43, 0.40, and 0.16, which show the reduced importance of burn severity to GL. The importance scores for GPP and ET represented resilience show variations. However, the overall conclusion regarding the contributions of these three metrics to resilience values remains consistent across the VTs. The train and test scores of different resilience values are included in Figure S11. The median $R^2$ scores for train and test datasets over 100 iterations

ranged between 0.68–0.71 and 0.62–0.67, 0.61–0.66 and 0.54–0.61, and 0.57–0.68 and 0.57–0.64 for LAI, ET and GPP, respectively for the three VTs. As the median $R^2$ scores for train and test datasets are close, it suggests the model is not significantly overfitting and learning the underlying patterns in the dataset (Yang et al., 2024; Yildirim et al. 2021).

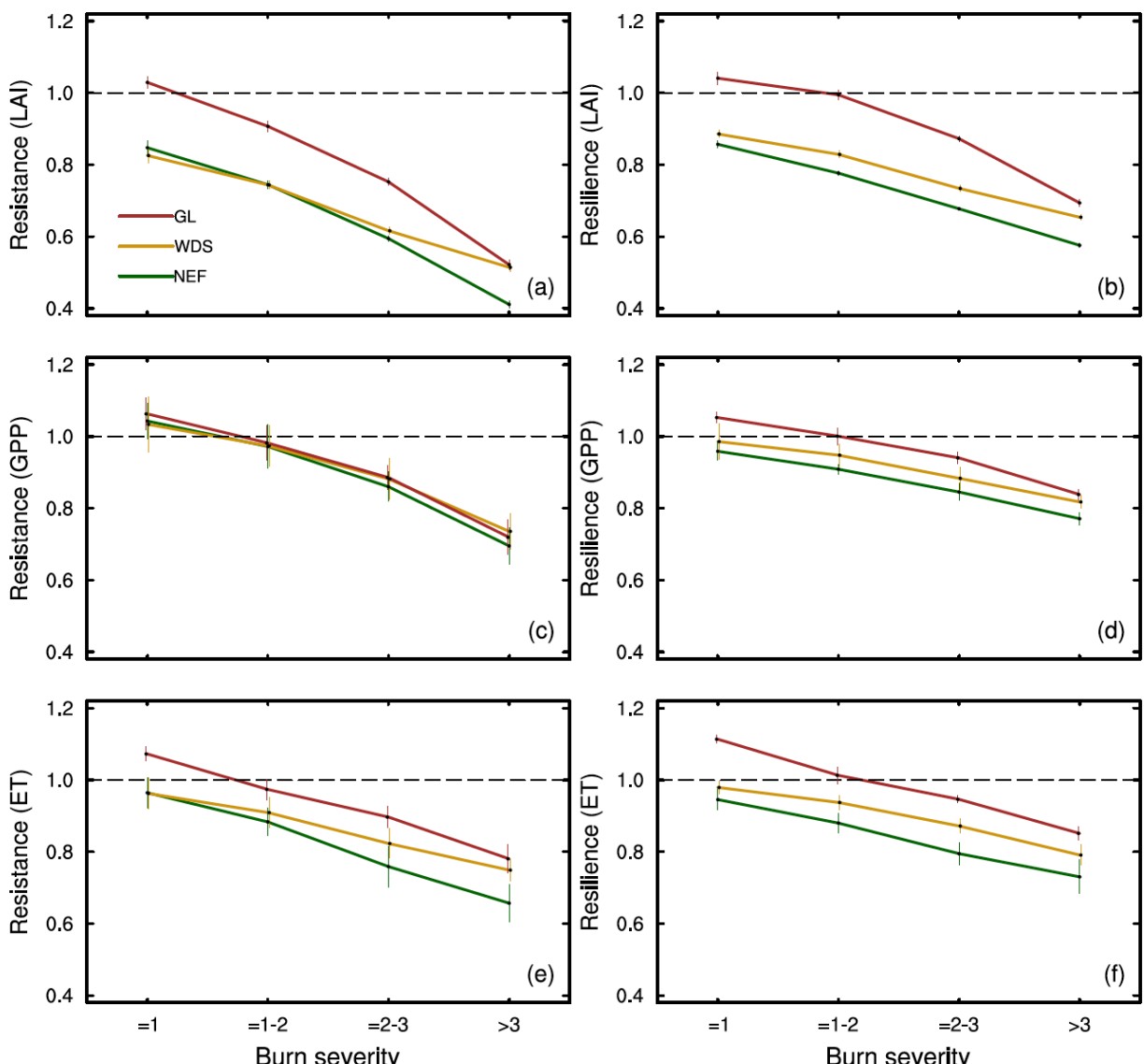

**Figure 2.** LAI (a) resistance and (b) resilience, GPP (c) resistance and (d) resilience, and ET (e) resistance and (f) resilience in needleleaf evergreen forests (NEF), woody savannas (WDS), and grasslands (GL) with different burn severities (Table S2). Resistance is calculated using the 2016 data. The 2016 resistance are the same as those shown for $S_{burn}$>3 in Figure 3, and are retained here to show the trends. The resilience calculation uses the mean of 2017–2020. The standard errors are included to represent the data variability across data pixels.

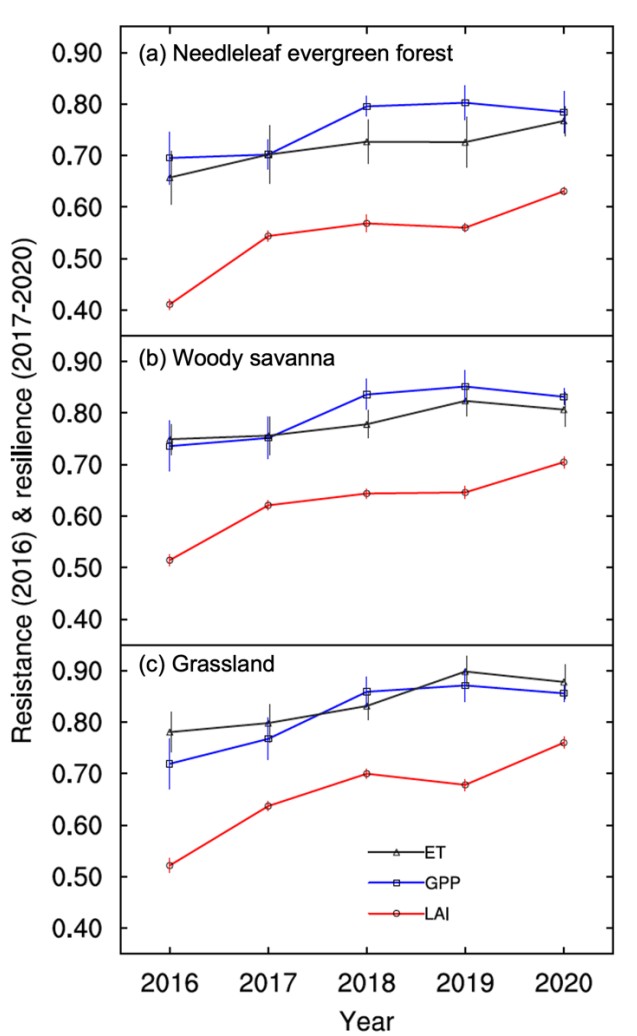

**Figure 3.** LAI, GPP, and ET temporal trends wildfire resistance (2016) and resilience (years 2017–2020) for (a) needleleaf evergreen forests, (b) woody savannas, and (c) grasslands with burn severity ($S_{burn}$) larger than 3. The standard errors are included to represent the data variability across data pixels.

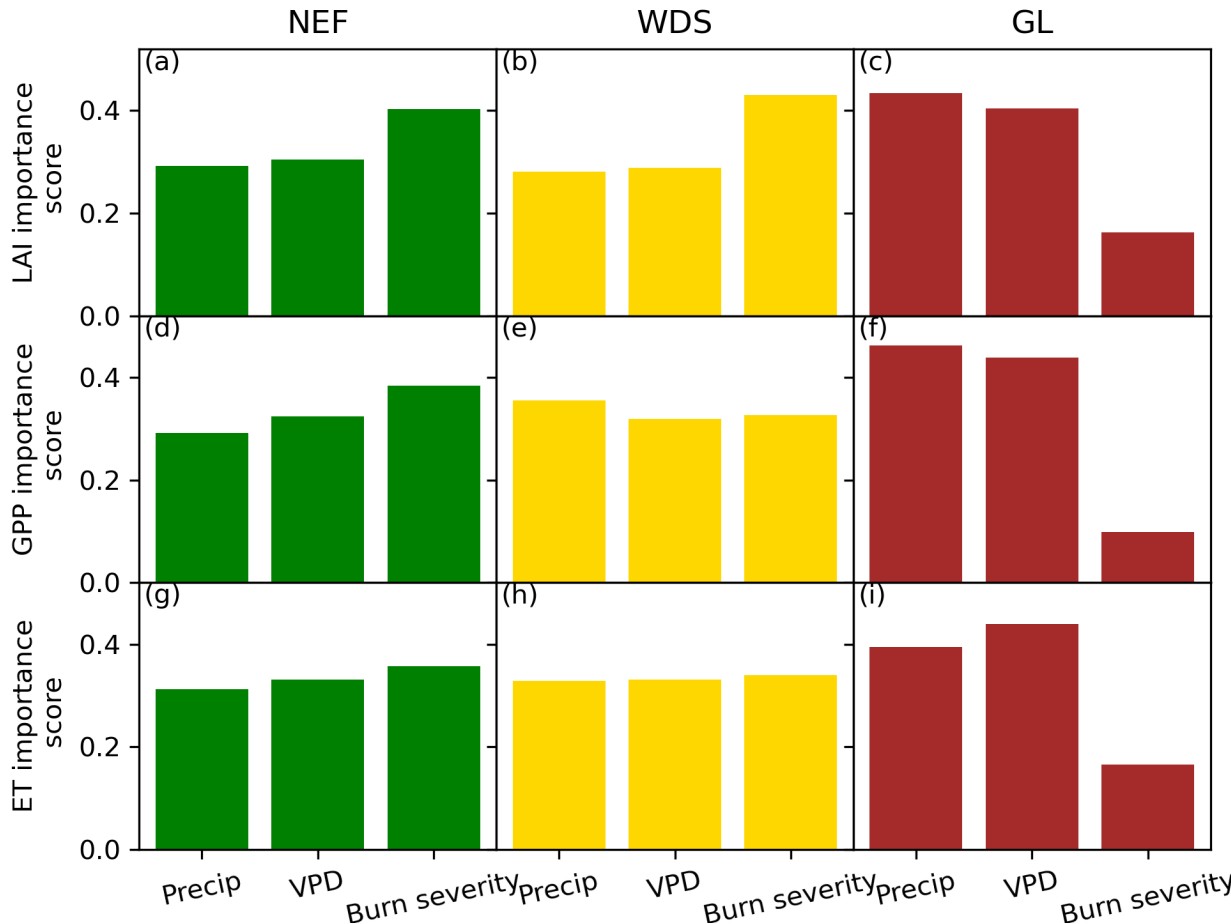

**Figure 4.** The feature importance of precipitation, VPD, and burn severity to resilience values in 2020 for LAI (a–c, NEF, WDS, and GL, respectively), GPP (d–f), and ET (g–i).

## 4.    Discussions

This study examines the immediate impacts and subsequent recovery of vegetation to 138 2015 CRB wildfires with multiple burn severity levels by using remotely sensed metrics of LAI, GPP, and ET within a formal resistance and resilience framework (DeSoto et al., 2020). The random forest feature importance algorithm is used to quantify the contributions of different factors, i.e., precipitation, VPD, burn severity, to resilience. This study quantitatively assesses the post-fire resistance and resilience determined by different MTBS burn severity categories in

different VTs through simultaneously using the three MODIS products (i.e., LAI, GPP, and ET). Overall, resistance and resilience reductions are closely related to burn severity increase, which matters more to the forest VT. Resilience is influenced by precipitation, VPD, and burn severity across all three VTs; burn severity plays a more important role in the resilience of forest and savanna VTs, while precipitation and VPD are more essential to the resilience of grassland.

## 4.1 Burn severity controls on ecosystem recovery

In all the studied VTs with burn severity larger than 3, the ecosystem status (i.e., LAI) and fluxes (i.e., GPP and ET) did not recover to pre-fire conditions within the post-fire study period (Figure 1). Figure 2 also shows that the increase of burn severity will reduce resistance and resilience across all the VTs, implying the lengthened recovery time of vegetation structure and fluxes. Among the three VTs, resistance and resilience are highest in grasslands, intermediate in woody savanna, and lowest in needleleaf forests. By exploring plant diversity and measured productivity across a variety of grassland sites in Europe and North America, Isbell et al. (2015) demonstrated full recovery one year after some drought disturbances. These trends can be attributed to ecosystems' evolution to tolerate various disturbance, where grasslands are adapted to more frequent fires and droughts in part through resprouting from their extensive root systems and can regrow leaf area far more rapidly than forests (Ratajczak et al., 2014; Isbell et al., 2015). These results support our first and second hypotheses that (1) higher burn severity results in lower resistance and resilience across all VTs, and (2) with the same burn severity, resistance and resilience are highest in grasslands, intermediate in savanna, and lowest in forests. Here, we estimate resistance by using the values of these three metrics in the first post-fire year (i.e., 2016) for all the VTs. When burn severity is 1, some ecosystems, such as grasslands, can recover rapidly,

and the values in 2016 may exceed those in 2015 or in 2011–2014 (see Figure S7). These variation (Figures S7c, S7f, and S7i) are primarily influenced by the minimum disturbance intensity, which could facilitate a quick recovery under favorable climate conditions.

It suggested by previous studies that severe fires can induce VT changes and ecosystem degradation (e.g., Karavani et al., 2016; Kumar et al., 2024). To further investigate the role of VT

changes in ecosystem resilience over CRB, we apply the burn severity map (Figure S2c) to both the 2014 and 2020 VT maps (i.e., the regions without fire disturbance are excluded), and the results show that among the 67090 fire disturbed data pixels in CRB, 13363 (20%) NEF, 7832 (12%) WDS , and 706 (1%) GL pixels experience VT changes (Figures 5 and Figure S12). In other words, most of the fire disturbed pixels with VT changes contain woodlands. This finding further justified

that grasslands are better adapted to wildfires (Isbell et al., 2015). Figure 2 shows that the resilience values of WDS are comparable to those of GL at high burn severities (i.e., $S_{burn}>3$), while at relatively low severities, they align more closely with those of NEF. Since WDS is an ecosystem that combines elements of woodland and grassland, this result implies that the grass component can recover quickly even under high burn severity, despite the potential damage to both trees and

grass. Furthermore, it is shown by previous studies that it can take decades to over a hundred years for the recovery of needleleaf trees (e.g., Turner et al., 2019). Thus, the existing observational time frame (e.g., till 2024) is not long enough for quantifying the forest recovery, and whether trees can grow back in these NEF and WDS is uncertain. The low resilience values in Figures 2, especially under high burn severity categories, could also be associated with fire induced VT shifts (e.g., tree

to grass). One possible scenario is that in the next a few decades, NEF will regenerate, and the dominance of grass will decrease with the reestablishment of trees, which will compete with grassland for light and nutrients. This assumption can be tested by dynamic global vegetation

models (DGVM), such as the Energy Exascale Earth System Model Land Model (ELM-FATES; Koven et al., 2020), which can be used to perform long-term simulations. Thus, using DGVMs to
further understand the post-fire vegetation dynamics in CRB and other regions with similar climate conditions could be our future research directions.

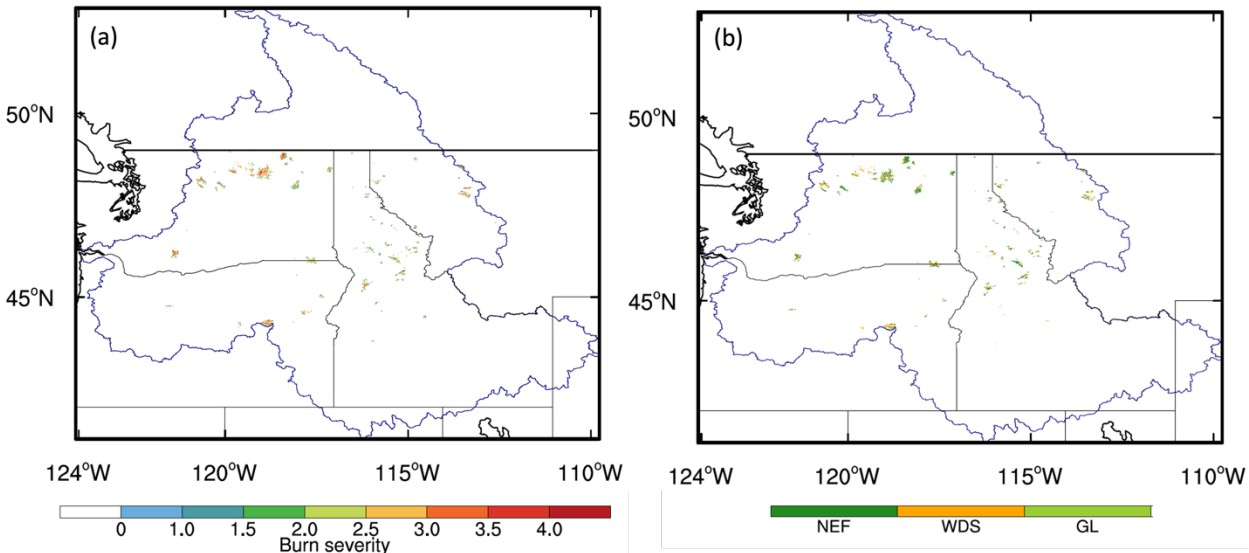

**Figure 5.** The spatial distribution of (a) the burn severity of data pixels experienced the 2015 fires
and (b) the pre-fire (i.e., 2014) VTs in the corresponding pixels of (a). NEF, WDS, and GL represent needleleaf evergreen forest, woody savanna, and grassland, respectively.

**4.2 The interactions between the carbon pools and ecosystem fluxes**

In all the resistance and resilience calculations based on different MODIS products, LAI
has the lowest resistance and resilience values, whereas GPP and ET have similar values (Figure 3). In other words, structure (i.e., LAI, a proxy of canopy biomass) has lower resistance and resilience than ecosystem fluxes (i.e., GPP and ET). The results are consistent with the previous studies showing that forests tend to increase stomatal conductance and hydraulic efficiency,

promoting the return of tree-scale transpiration after fires (Nolan et al., 2014). Cooper et al. (2019)
also show the enhanced transpiration rates for forests with moderate burn severity. All these findings support the relatively quicker recovery of GPP and ET than that of LAI.

GPP is an indicator of "ecosystem-wellness" (Frazier et al., 2013) and ET is a water budget component mostly sensitive to vegetation changes (DeBano et al., 1998). The resistance and resilience value differences (Figures 1, 2, and 3) also show the different recovery features of these two variables in various burn severities and VTs. We acknowledge that the MODIS GPP and ET products use the MODIS LAI and fraction of Photosynthetically Active Radiation (fPAR) and the same meteorological data for their estimates (Running et al., 2004). Specifically, the MODIS GPP algorithm uses (1) the MODIS fraction of photosynthetically active radiation (FPAR) retrieved at the same time with LAI, (2) the MODIS daily mean Photosynthetically Activate Radiation (PAR), (3) the biome-specific radiation use efficiency parameters that are extracted from the Biome Properties Look-Up Table (BPLUT) and adjusted by the temperature and VPD scalars, to estimate GPP (Running and Zhao, 2019). The estimates of MODIS ET use the Penman-Monteith equation, which considers many environmental factors including radiation components, air temperature, and relative humidity. Here, LAI is used to estimate wet canopy resistance to sensible heat and resistance to latent heat transfer, which will be used to calculate evaporation on wet canopy surface (Running et al., 2019). Thus, the recovery of GPP and ET are also affected by post-fire environmental factors, which are not responding to burn severity and have large daily, seasonal, and interannual variabilities. The similar responses of GPP and ET to fires could be associated with the tightly coupling between these two fluxes, which is governed by stomatal conductance. Stomatal conductance regulates both photosynthesis and transpiration (Knaue et al., 2020; Stoy et al., 2019), and the correlations between GPP and ET (Running et al., 2004). Partitioning the

contributions of GPP and ET coupling, as well as the methods used to derive GPP and ET data, is beyond the scope of this study. Here, our third hypothesis is supported, and it suggests that resistance and resilience post-disturbance is highest for GPP and ET, and lowest for LAI across all VTs.

**4.3 The environmental controls on resilience**

It is shown by Lu and Yan (2023) that GPP can decrease with the increase of VPD in a grassland ecosystem, where this VPD limitation on GPP is not obviously shown in a nearby forest ecosystem. Since both precipitation and VPD largely influence vegetation growth in the semi-arid region, we explicitly test the fourth hypothesis to further understand the contributions of environmental factors (i.e., precipitation and VPD) and burn severity in determining the post-fire ecosystem recovery between VT. The random forest feature importance study reveals that precipitation, VPD, and burn severity have various impacts on resilience represented by different variables across VTs. Even though burn severity is less important to grassland resilience (Figure 4), the forest and savanna VTs show a stronger influence of burn severity on resilience in terms of LAI. Together, these results point to the interaction of precipitation, VPD, and burn severity in regulating ecosystem resilience, and the higher and longer lasting impacts of wildfires on VTs with higher biomass. The results also show that our fourth hypothesis, anticipating higher importance of precipitation and VPD to resilience in grassland than in other VTs, is testable. The post-disturbance biotic factor determined slow recovery of the forest ecosystems is also identified by Shi et al. (2017), who perform numerical simulations based on the 2005 Amazonian drought with the Community Land Model (CLM), revealing the limited influence of environmental factors to the forest recovery.

The random forest feature importance study implies that hydraulics are influenced almost equally by water supply (i.e., precipitation) and demand (i.e., VPD). Here, Figures 1 and 3 show similar recovery trends across NEF, WDS, and GL, while Figure 4 implies various importance scores of different factors in determining water dynamics and burn severity. We notice the differences in precipitation and temperature among these three VTs (Figure S4), with relatively

drier and hotter conditions in GL, which underscores the role of climate in shaping the distribution of different VTs (Mather et al., 1968). In other words, the importance scores highlight the roles of different factors in recovery across VTs, which adapt to specific environmental conditions (Figures S1, S3, and S4) and exhibit their respective recovery rates. Advanced studies are needed to investigate the varied impacts of precipitation and VPD on resilience in different ecosystems with

various types of disturbance (e.g., droughts and heatwaves), which will further imply ecosystem recovery capacity and functionality shifts due to disturbance and is beyond the scope of this study. In addition, the post-fire interactions between LAI, GPP, and ET, and the sensitivity of these variables to burn severity and environmental factors during the recovery process can be further evaluated by using ESMs. Research toward this direction can be used to guide model's

parameterization and enhance process-based representation in reasonably characterizing impacts of fires in different ecosystem types in ESMs.

**4.4 Data uncertainty and application limitations**

       It is shown by previous studies that spectral observations of forests' canopy characteristics

(e.g., leaf area) tend to be biased, resulting from clouds and aerosol on the measurement pathways (Asner and Alencar, 2010; Samanta et al., 2012; Xu et al., 2011). Therefore, the application of this research framework to other regions with fire disturbance, especially in the tropics with density

vegetation coverages, is limited by the observational capacity of spectral-based measurements. This also implies that intensified airborne measurements and Lidar measurements can be

extremely useful for enhancing the fundamental understanding of ecosystem processes after disturbances.

## 5. Conclusions

The results of this study suggest the decrease of resistance and resilience to burn severity

increases and the higher resilience of grassland compared to forest and savanna VTs, which is largely regulated by precipitation and VPD during the recovery processes. Our research affirms the findings from plot-based measurements and shows a strong potential for using satellite observations to investigate ecohydrological processes and resistance and resilience to different types of disturbance in regions with reasonable data quality controls (i.e., with the remote-sensing

quality control flags considered). The data revealed recovery feature differences between LAI, GPP, and ET and their variations between VTs that determined by varied burn severity can be largely used to interpret fire-induced the carbon and water cycle changes in terrestrial biosphere models (TBMs) and enhance models' parameterization. The findings obtained from the random forest feature importance study can also be used to guide sensitivity tests (e.g., precipitation

perturbation tests) and advance physical based understanding in TBMs.

With anticipated hotter and drier fire seasons with extended duration in the Pacific Northwest according to future climate projections (Wimberly et al., 2014), we expect that fire frequency and burn severity of wildfires will be increasing with the changing climate patterns. This study implies that with these changes, some ecosystems may need long time frames to achieve

full recovery. This prolonged recovery could keep carbon stocks at relatively low levels for

decades to a century (Turner et al., 2019) and affect the ecosystem function and ecosystem–atmosphere interactions (Harris et al., 2016). Thus, both data-oriented studies and the enhanced model capacity in reasonably predicting fire frequency and burn severity and properly characterizing impacts of fires in different ecosystem types are essential to the research of the

carbon cycle, ecosystem functioning, and climate change.

## 6. Data availability statement

The data are publicly available at the ESS-DIVE repository under the Creative Commons Attribution 4.0 International License using [doi:10.15485/250704](doi:10.15485/250704)

## 7. Author contribution

MS, NM, and XC were responsible for conceptualization. MS and NM designed the study and wrote the first draft of the paper. HH prepared and processed the burn severity data, and LL provided the vegetation type data. MS worked on the remote-sensing data analysis and visualization, and FS performed the random forest feature importance tests. XC was responsible

for funding acquisition. All the authors contributed to manuscript revising.

## 8. Competing Interests

The contact author has declared that none of the authors has any competing interests.

## 9. Acknowledgements

We would like to sincerely thank the two anonymous reviewers for their valuable comments and constructive feedback, which greatly improved the quality of this work. This research was supported by the U.S. Department of Energy (DOE), Office of Science (SC) Biological and Environmental Research (BER) program, as part of BER's Environmental System Science (ESS)

program. This contribution originates from the River Corridor Scientific Focus Area (SFA) and the Interoperable Design of Extreme-scale Application Software (IDEAS)-Watershed Project at Pacific Northwest National Laboratory (PNNL). PNNL is operated for DOE by Battelle Memorial Institute under contract DE-AC05-76RL01830. This paper describes objective technical results

and analysis. Any subjective views or opinions that might be expressed in the paper do not

necessarily represent the views of the U.S. Department of Energy or the United States Government.

This research used resources of the National Energy Research Scientific Computing Center, a DOE

Office of Science User Facility supported by the Office of Science of the U.S. Department of

Energy under Contract No. DE-AC02-05CH11231, using NERSC award BER-ERCAP0023098.

**10. Financial support**

This research was supported by the U.S. Department of Energy (DOE), Office of Science (SC)

Biological and Environmental Research (BER) program, as part of BER's Environmental System

Science (ESS) program.

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
