# Peer review of "Ecosystem leaf area, gross primary production, and evapotranspiration"

_EGUsphere, 2024_

## Author Comment (AC1)

**Reviewer #1**

General comments:
The paper "Ecosystem leaf area, gross primary production, and Evapotranspiration Responses to Wildfire in the Columbia River Basin" examines how different vegetation types recovered after a severe fire season during 2015, looking at MODIS LAI, GPP and ET products combined with the MBTS burn serenity classification data. The authors targeted concepts of "resistance" and "resilience" by looking at the difference in LAI, GPP, and ET in years following 2015 compared to the interannual variability of years prior to 2015. The authors then use a random forest feature importance to determine if vapor pressure deficit, precipitation, or burn severity were most important in determining how resilient a vegetation type was. They found that grasses were more resistant and resilient than savannahs and needleleaf evergreen forests, and that LAI was more responsive to burn severity than GPP and ET.

I found this paper to address important scientific concepts. The authors' application of concepts of resilience and resistance was an important and interesting lens to explore how disturbances affect landscapes. I also found the paper to have some major flaws, specifically pertaining to how the authors document uncertainty within their analyses, and the independence of the MODIS LAI, GPP, and ET products. I identified minor flaws in some elements of how the analysis was scoped and discussed.

Notably, the materials I received for review had no supplemental materials, but the paper makes reference in the text to several supplemental elements. The text will need to be edited substantially if the supplement was intentionally excluded.

**Major comments:**
This paper lacks discussion of uncertainty in interpreting the results presented. For example, in Figure 2 the only delineation of uncertainty is that of interannual variability. It's unclear if that interannual variability is mean to be interpreted as the threshold for "significant difference", and if that is the case it is in contradiction with some of the writing (see comments for line 355). A single number to describe a year gave me pause as a reviewer. Similarly, Figures 3 and figures 4 present single-point comparative data with nothing to contextualize the uncertainty around that single point. This is particularly relevant for figures 3 and 4, because the range in inter-annual resistance and resilience presented in figure 4 is sometimes larger than the differences in resistance and resilience between different vegetation-types. Interpreting if the differences presented are meaningful is left to the reader, undercutting interesting patterns. In the text, discussion of uncertainty only acknowledges limitations of remotely-sensed LAI, and does not discuss the specific uncertainties and limitations unique to the paper itself. While more appropriate to the methods section, the text also does not discuss spatial uncertainty.

**Reply:** We noticed the relatively low values in both precipitation and temperature in 2019. It is suggested by previous studies that plants adjust the dynamics of stomata to optimally satisfy the trade-off between the amount of carbon assimilated and the amount of water transpired (Cowan and Farquhar, 1977). Thus, vegetation growth and potential photosynthetic carbon (C) uptake or

gross primary production (GPP) are closely related to evapotranspiration (ET; Brümmer et al., 2012), and investigating ET and water use efficiency (WUE; the ratio of GPP to ET) changes is essential to the understanding of their roles in determining the carbon and water variabilities. The variations of ET are usually restricted by both temperature and soil water availability in temperate and high latitudes (Brümmer et al., 2012). For these reasons, we plotted the WUE changes to further understand the carbon and water interactions as a result of the interannual variability of precipitation and temperature (Figure R1).

In 2019, both precipitation and temperature are lower than those in other years during the recovery period (i.e., 2016–2020). The low temperature can reduce ET (e.g., soil evaporation) and maintain relatively high WUE and GPP values as long as the soil water availability is sufficient to support the growth. Compared to other years, 2017 and 2020 have relatively higher precipitation and surface air temperature values, and the annual WUE is lower than that in other years, implying the relative high levels of water usage to maintain the growth. These trends indicate that the interactions between ET and GPP are non-linear and high precipitation and surface air temperature values are not always facilitating high GPP values due to the close connections between ET and GPP (Figure R1). The investigation between ET and GPP across ecosystems is out of the scope of this study.

[Figure]

**Figure R1.** The growing season water use efficiency (WUE) over the three vegetation types with burn severity >3 during 2011–2020.

To address the reviewer's concern regarding the spatial specific uncertainties and spatial variations of the data, we calculated the standard error by using the LAI, GPP, and ET values in data pixels of each vegetation type, and added error bars in Figures 1–3 (newly numbered figures). The implication of the error bars in Figures 1 and S7 is added in Section 3.2. The interannual variation and the variations between vegetation types in terms of resistance and resilience are discussed in the second paragraph of Section 3.3. Overall, the standard errors indicate that LAI

and it represented resilience have relatively smaller spatial variations than those of GPP and ET. Woodlands (i.e., NEF and WDS) can have similar resilience values when they are disturbed by severe fires (i.e., $S_{burn} > 3$).

Second, it is unclear if the MODIS LAI, GPP, and ET are independent enough to draw separate conclusions. The authors wisely acknowledge in line 384 that the MODIS GPP and ET products use MODIS LAI as an input, along with meteorological data. However, this does call into question if the finding that LAI responds more to burn severity than GPP or ET, because GPP and ET definitionally contain the same information as the LAI product, plus other variables that don't respond to burn severity. If the GPP and ET product contain information that could respond to burn severity beyond LAI, then that needs to be explicitly stated.

**Reply:** We agree with the reviewer that GPP and ET indicate some changes that are not closely related to burn severity. LAI is obtained from the MODIS LAI/fraction of photosynthetically active radiation (FPAR) algorithm that consists of a main Look-up-Table (LUT) based procedure, exploiting the spectral information content of the MODIS red and near infrared surface reflectance. For MODIS LAI/FPAR, there is also a back-up algorithm that uses empirical relationships between Normalized Difference Vegetation Index (NDVI) and canopy LAI and FPAR. The estimates of MODIS GPP use the (1) MODIS FPAR, (2) the GMAO/NASA provided photosynthetically active radiation (PAR), and (3) the biome-specific radiation use efficiency parameters that are extracted from the Biome Properties Look-Up Table (BPLUT) and updated by the temperature and vapor pressure deficit scalars. The estimates of MODIS ET use the Penman-Monteith equation, which considers many environmental factors including radiation components, air temperature, and relative humidity. LAI is used to estimate wet canopy resistance to sensible heat and resistance to latent heat transfer, which will be used to calculate evaporation on wet canopy surface. In all the calculations of GPP and ET, the environmental factors will not be responding to burn severity, especially during the recovery years, and have large daily, seasonal, and interannual variabilities. We added the following paragraph in the second paragraph of Section 4.2:

*"Specifically, the MODIS GPP algorithm uses (1) the MODIS fraction of photosynthetically active radiation (FPAR) retrieved at the same time with LAI, (2) the MODIS daily mean Photosynthetically Available Radiation (PAR), (3) the biome-specific radiation use efficiency parameters that are extracted from the Biome Properties Look-Up Table (BPLUT) and adjusted by the temperature and vapor pressure deficit scalars, to estimate GPP (Running and Zhao, 2019). The estimates of MODIS ET use the Penman-Monteith equation, which considers many environmental factors including radiation components, air temperature, and relative humidity. Here, LAI is used to estimate wet canopy resistance to sensible heat and resistance to latent heat transfer, which will be used to calculate evaporation on wet canopy surface (Running et al., 2019). Thus, the recovery of GPP and ET are also affected by post-fire environmental factors, which are not responding to burn severity and have large daily, seasonal, and interannual variabilities."*

**Minor comments:**

Methods: Grasses could grow back faster than the "resilience"/ "resistance" increment of 1 year. Is it appropriate to consider grasses' "resistance" in the first year post fire?

**Reply:** Figures 1 and S7 show that only when the burn severity is 1, the LAI, GPP, and ET values in 2016 could be higher than that in 2015 or in 2011–2014. In other cases, the values of these three metrics in 2016 are always the smallest, indicating that only when the burn severity is minimum, grasses and other ecosystem types can have a faster growth and recovery to the pre-fire conditions in the first year after fire. That is the higher the burn severity, the lower the values of these three metrics in the first year after fire. Here, we are quantifying resistance and recovery capacity, so use the lowest values that can represent the damage intensity (largest reduction of these variables) given that "*fires happened mid-way through the growing season (Figure S4)*" of 2015 and "*the 2015 values include both pre- and post-fire*" conditions (See Section 2.5). Overall, we comprehensively evaluated different factors and decided to use the values first year after fire for all ecosystem types. We also added the following sentence by the end of Section 4.1:

> "*Here, we estimate resistance by using the values of these three metrics in the first post-fire year (i.e., 2016) for all the VTs. When burn severity is 1, some ecosystems, such as grasslands, can recover rapidly, resulting in that values in 2016 may exceed those in 2015 or in 2011–2014 (see Figure S7). These variation (Figures S7c, S7f, and S7i) are primarily influenced by the minimum disturbance intensity, which could facilitate a quick recovery under favorable climate conditions.*"

Line 85- Citation how LAI related to prediction uncertainties of earth system models?

**Reply:** The citation is added.

Line 160-164 – This seems to contradict methods described in lines 193-195

**Reply:** We used the method in lines 193–195, and we updated 160-164 to avoid the confusion as "*We use the dominant VT of each fire event, defined as the VT whose area accounts for more than 50% of burned area for that event, to identify the representative landscape type of each fire event (Figure S2b). This analysis aims to comprehend which VT(s) are predominantly affected by fire across the CRB (Figure S2b). However, to precisely estimate the resistance and resilience of different VTs, we explicitly consider the VTs and their changes characterized by LAI, GPP, and ET in each data pixel of each fire event (see more details in Section 2.4).*"

Line 186 - Difficulty understanding the sentence beginning with "Specifically…" What information is applied? Is this referring to MODIS pixels that span multiple fires, or each fire-MODIS pair?

**Reply:** This sentence is updated as "*Specifically, within each MTBS fire boundary, the MODIS VT information for each data pixel is used to derive different VT determined LAI, GPP, and ET changes in the corresponding MODIS data pixels (Table S1).*"

Line 195 - Table 1 seems more appropriate to a supplement. Perhaps a column indicating the years used in this study?

**Reply:** Table 1 is moved to Supplement, and an additional column is added.

Line 245 - 248: Citation/discussion for why 100 trained models were sufficient.

**Reply:** We added citations regarding the 100 trained time simulations. We also performed new simulations by adding the training times to 500, and did not see any changes that affect our conclusions. Thus, we decided keeping the 100 training times in our study.

[Figure]

**Figure R2.** The new simulation with 500 training times. The $R^2$ scores for the random forest model on train and test datasets for the (a) needleleaf evergreen forest (NEF), (b) woody savanna (WDS), and (c) grassland (GL) VTs represented by LAI, the (d) NEF, (e) WDS, and (f) GL VTs represented by GPP, and for (g) NEF, (h) WDS, and (i) GL VTs represented by ET.

Lines 250-258: This paragraph reads as more relevant to the methods section, and the writing shifts from present tense to past tense.

**Reply:** The sentence "*During the fire season of 2015, 138 fire events are identified. We also remove the areas that experienced fire in 2011–2014 or in 2016–2020, thus our resistance and resilience*

*calculations are not confounded by repeat fires.*" is moved to Section 2.2, and this paragraph is updated as "*The MTBS and VT based analysis shows that August is the month with the highest fire frequency in 2015 with 91 fire events (Figure S2a), where NEF experiences 42, WDS experiences 27, and grassland experiences 67 fire events, respectively (Figures S1 and S2b). There are two fire events in croplands, which were excluded from further analysis.*"

Figure 1 - Also seems more appropriate for a supplement. More discussion/ visual delineation of uncertainty would make this a more powerful figure. For example, why not include error bars for the years themselves? Finally, while the text claims climate variations in years (is that distinct from interanual variability?) are not confounding, 2019 seems to be different than other years compared to the interannual variability of 2001-2014 in precipitation. This is not discussed, and specifically how years are not significantly different is not defined.
**Reply:** Figure 1 is moved to SI.

Line 355 - Seems to contradict Figure 2 – 2019 is higher than 2020 in GPP. Also, if "Recover" is defined as reaching the same value within the variability of 2011-2014, then many metrics did not "recover" at all.
**Reply:** This comment is addressed earlier.

Line 321-322 Cite statement about overfitting.
**Reply:** Yang et al. (2024) and Yildirim et al. (2021) are cited.

443 - The connection between this research and afforestation and sustainable agriculture is unclear in this current writing.
**Reply:** This sentence is removed from the manuscript.

Conclusions: Potential uses for this research for the calibration of ESM is discussed for the first time in the conclusions.
**Reply:** We also add discussions of the importance and implications of using ESMs in the last sentence of Section 4.3.

**Technical Corrections:**
Line 209: - Additional period.
**Reply:** The period is deleted.

Line 278: "S" denoting severity and "S" denoting supplement both is confusing.
**Reply:** "S" is updated to "$S_{burn}$" to represent burn severity in Figures in the main context (e.g., Figure 2) and in Supplementary.

Line 316 important scores -> importance scores
Reply: Updated.

364 - fully – full

Reply: Updated.

395- interacting -> interaction
**Reply:** Updated.

425 "findings that obtained with"  -> findings from, "potential of" -> "potential for"
**Reply:** Updated.

428 - unclear what is meant by "reasonable data quality controls"
**Reply:** This sentence is updated as "*Our research affirms the findings from plot-based measurements and shows a strong potential for using satellite observations to investigate ecohydrological processes and resistance and resilience to different types of disturbance in regions with reasonable data quality controls (i.e., with the remote-sensing quality control flags considered).*"

**Reference:**

Brümmer, C., Black, T. A., Jassal, R. S., Grant, N. J., Spittlehouse, D. L., Chen, B., ... & Wofsy, S. C. (2012). How climate and vegetation type influence evapotranspiration and water use efficiency in Canadian forest, peatland and grassland ecosystems. *Agricultural and Forest Meteorology*, *153*, 14-30.

Cowan, I. R., & GD, F. (1977). Stomatal function in relation to leaf metabolism and environment.

Yang, B., Heagy, L. J., Morgenroth, J., & Elmo, D. (2024). Algorithmic Geology: Tackling Methodological Challenges in Applying Machine Learning to Rock Engineering. Geosciences, 14(3), 67.

Yildirim, M. O., Gok, E. C., Hemasiri, N. H., Eren, E., Kazim, S., Oksuz, A. U., & Ahmad, S. (2021). A machine learning approach for metal oxide based polymer composites as charge selective layers in perovskite solar cells. ChemPlusChem, 86(5), 785-793.

---

## Author Comment (AC2)

**Reviewer #2**

In "Ecosystem leaf area, gross primary productions, and evapotranspiration responses to wildfire in the Columbia River Basin" the paper compares metrics of resistance and resilience of the MODIS-derived products mentioned in title across 3 primary vegetation types that underwent different burn severities from 138 fires in 2015. Resistance is calculated as the ratio of the 2016 annual maximum value for LAI, GPP, and ET to the average annual max value from the 4 years preceding the fire (2011-2014). Resilience is ratio of the average annual max values of LAI, ET, and GPP during 4 post-fire years (2017-2020) to the average annual max values during the 4 pre-fire years. A random forest approach is used to compare the influence of precipitation, vapor pressure deficit and burn severity on resilience of the three vegetation types (needleleaf evergreen forest, woody savanna, and grasslands). The paper concludes that 1. increased burn severity decreases resistance and resilience of LAI, GPP, and ET across all vegetation types, 2. LAI resistance and resilience is the most impacted by burn severity, 3. grassland LAI, GPP and ET are more resilient and resistant to wildfire than that of evergreen forests and savannas, and 4. burn severity is the primary driver of resilience in evergreen forests and savannas, while VPD and precipitation determine grassland resilience.

This paper's methods and results are thought provoking and likely of interest to many. In addition to sharing concerns regarding the uncertainty issues addressed by the first reviewer, I have identified other limitations that should be addressed. Namely, the resistance and resilience metrics in this study are produced using different data than in the key cited paper (DeSoto et al., 2020) and thus there must be clear differences in the interpretation of the results.

The short post-fire study period undermines discussion related to resilience. This paper claims that low resilience values relate to low chances of full recovery (438-439). However, a 4-year post-fire study period is too short to assess the likelihood of recovery when forest recovery can take decades. If the authors decided to retain this resilience metric, the interpretation needs to appropriately acknowledge and match the limitations of the method.
**Reply:** This sentence is updated as "*This study implies that with these changes, some ecosystems may need a long time frame to achieve full recovery. This prolonged recovery could keep carbon stocks at relatively low levels for decades to a century (Turner et al., 2019) and affect the ecosystem function and ecosystem–atmosphere interactions (Harris et al., 2016).*"

Additionally, the authors do not use DeSoto's ecosystem recovery metric. At the very least, there should be discussion of why that is not used since they otherwise rely on DeSoto's framework. DeSoto et al. 2020 define recovery as the ratio between the post disturbance value and the value during the disturbance, and thus resilience (post disturbance/pre disturbance) is a product of both resistance (disturbance/pre disturbance) and recovery (post disturbance/ disturbance). Based on figure 2, recovery of LAI, GPP, and ET appear quite similar across the three vegetation types following severe burn. Thus, observed differences in resilience are driven primarily by differences in resistance as opposed to recovery. I believe that it is key that this paper properly discuss the nuanced differences in interpreting recovery (seemingly similar across these ecosystems) and resilience and resistance (seemingly contrasting).

**Reply:** To simplify our analysis, we use resistance and resilience to represent disturbance intensity (i.e., disturbance/pre-disturbance) and the capacity of a system to recover (i.e., post-disturbance/pre-disturbance), respectively. The equations of these two metrics are following Eqs. (1) and (3) in DeSoto et al. (2020). Essentially, the product of resistance (disturbance/pre-disturbance) and recovery (post-disturbance/disturbance) cancels out the "disturbance" term—defined as the value during disturbance. Figure 2 (now Figure 1) focuses solely on resistance (i.e., values in 2016) and resilience (values in 2017 and after).

The reviewer commented that "*Thus, observed differences in resilience are driven primarily by differences in resistance as opposed to recovery*". However, based on Eq (2) in this paper or Eq (3) in DeSoto et al. (2020),

$$\text{Resilience} = \frac{\overline{A}_{2017-2020}}{\overline{A}_{2011-2014}}$$

The status in 2017 and after could be closely related to the status in 2016 (i.e., the value during disturbance), and compared to resistance, the resilience values are determined by the post-disturbance values, implying the ecosystems' capacity to recover. On the other hand, "recovery" is represented as DeSoto et al. (2020):

$$\text{Recovery} = \frac{\overline{A}_{2017-2020}}{A_{2016}}$$

Based on Eqs. (1)–(3), the only difference between "resilience" and "recovery", as defined by DeSoto et al. (2020). is whether the pre-disturbance value or the value during disturbance is used as the denominator for the calculation. These three variables, resistance, resilience, and recovery, are not one determine another, but use different time frames, pre-disturbance—disturbance— post-disturbance, to characterize the system's responses to disturbance with different intensity. We also added the following description in Section 2.5:

"*DeSoto et al. (2020) define recovery as the condition following a disturbance compared to the condition in the year when the disturbance occurred, which can be represented by the following equation:*

$$Recovery = \frac{\bar{A}_{2017-2020}}{A_{2016}} \qquad (3)$$

*Thus, the primary distinction between resilience and recovery lies in the reference conditions: pre-fire versus the disturbed conditions. Resilience is defined as the capacity of a system to recover its structure and function after a disturbance (Holling, 1973). To ensure clarity in our discussions of recovery status, we use the term resilience throughout this study.*"

**Additional Major Comments:**

Burn severities are presented as a key driving factor but I could not find information (in either the article or SI) about the distribution of burn severities across the different primary vegetation types. This would provide useful context for figures 3 and 5. Also, there is no discussion concerning the issues with burn severity classifications in grasslands compared to forests. Are there

differences in classifications between vegetation types that readers should consider? Do MTBS algorithms and workflows perform similarly well in grasslands, savannas and forests?

**Reply:** Figures S1 provides detailed information of VTs and Figure S2c provides burn severity levels over the CRB. In addition, we add Figure S2d to further show the VTs in all the Figure S2c indicated fire scars. Typically, northern CRB is more prone to forest fires, whereas southern CRB is primarily affected by grassland fires. We also provide the fire types across different VTs (Figure S2b). We also add the description in the first paragraph of Section 3.1 as:

*"Generally, northern CRB experiences a higher incidence of forest fires, whereas southern CRB is mainly affected by grassland fires (Figures S1, S2c, and S2c)."*

The MTBS mapping is primarily based on the differenced normalized burn ratio (dNBR) algorithm and the LandSat imagery at the near-infrared and SWIR bands (Eidenshink et al., 2007). The same dNBR algorithm is applied to all vegetation types, and differenced NBR images (i.e., post-fire NBR subtracted from pre-fire NBR) are referred to as dNBR images. The differenced pre-fire and post-fire NBR images result in a fire-related change image that is classified into severity classes and provides an unbiased basis for analyzing additional fire effects. Thus, a same burn severity value in both a forest and a grassland system implies these two systems have the same range of changes (i.e., variations) in terms of NBR. The clarification is also added the following sentence in the second paragraph of Section 2.2:

*"The MTBS mapping primarily relies on the differenced normalized burn ratio (dNBR) algorithm and LandSat imagery in the near-infrared and SWIR bands (Eidenshink et al., 2007). The same dNBR algorithm is applied across all VTs. Differenced NBR images—where post-fire NBR is subtracted from pre-fire NBR—are known as dNBR images. These dNBR images illustrate fire-related changes, which are categorized into severity classes, providing an unbiased foundation for analyzing additional fire effects."*

Vegetation classifications are based on a 2015 MODIS dataset. For those areas that burned in 2015, do more recent land cover type datasets suggest a possible change in vegetation cover type? The authors do not discuss possible land cover conversion following different severity burns. Indeed, in figure 3 savannas seem to respond like grasslands at high severities and forests at low severities. While the post-fire study period is not long enough to expect full forest or savanna recovery from severe burn, more explicit discussion on conversion is warranted.

**Reply:** We compared the vegetation types (VTs) in 2014 (pre-fire) and 2020 (the fifth year after fire), and applied the burn severity map (with fire boundaries; Figure S2c) to the VT difference map to show the fire scars with VT changes. That is the data pixels without fire but with VT changes are excluded from this evaluation. This new comparison is included in Figure 5 and Discussed in Section 4.1. The results showed that there were 67090 data pixels in the Columbia River basin experienced the 2015 fires. Even though there are many grassland fire events (Figure S2b), 13363 (20%) needleleaf forest, 7832 (12%) woody savanna, and 706 (1%) grassland pixels experienced VT changes among all the fire disturbed pixels (Figure 5). Thus, the 2015 wildfires induced the degradation of VTs with woody components. As we discussed in the paper that grasslands are better adapted to wildfire (e.g., Isbell et al., 2015), this analysis further confirmed

the results of previous studies. We added the discussion in the last paragraph of Section 4.1 to further explain the resilience value variations between VTs.

**Technical corrections:**

61-62: change to: "greater reductions in ET to precipitation ratios (ET/P)"
**Reply:** Updated.

63: Is "respectively" needed?
**Reply:** "respectively" is deleted.

63-66: Please improve the wording of this sentence.
**Reply:** This sentence is updated as "*Through the use of the Moderate-resolution Imaging Spectroradiometer (MODIS) GPP product, a Hurricane Rita based study suggests the difference in GPP recovery rates and resilience index between different vegetation types (Frazier et al., 2013).*"

70: Start sentence with "There".
**Reply:** Updated.

83-85: Rearrange this sentence, possibly split into two sentences.
**Reply:** This sentence is updated as "*Furthermore, the influence of environmental factors on post-fire ecosystem recovery, as characterized by LAI, GPP, and ET, remains inadequately quantified. This lack of clarity is closely linked to the uncertainties in predictions made by Earth system models (ESMs; Lawrence et al., 2016).*"

93: Replace ", experienced" with "across"
**Reply:** Updated.

95 -105: It would be easier to read if the hypotheses were simply listed. The discussion of concerning results from previous studies may be better earlier in the introduction or later in the discussion section.
**Reply:** The hypotheses are directly listed, and the fifth paragraph in the Introduction and the first paragraph in Section 4.3 are reorganized.

108-113: The final part of the sentence about increasing wildfire severity in the CRB should be another sentence.
**Reply:** The final part is updated as "*Given that the impacts of wildfires are expected to intensify in CRB (Halofsky et al., 2020; Wimberly et al., 2014) and globally (Andela et al., 2017; Bowman et al., 2020., Jones et al., 2024), quantifying fire impacts is essential for both ecosystems and society. The research framework of this study can be broadly applied to quantify wildfire-induced ecosystem responses and evaluate the impacts of wildfires as revealed by different data products and represented by ESMs.*"

361: Including "(e.g., grasses)" is misleading. DeSoto et al. 2020 compares gymnosperm and angiosperm *tree* species, there is no discussion about grassland recovery.
**Reply:** This sentence was deleted.

116: Unnecessary first sentence.
**Reply:** Deleted

130: include "annual 500 m" before "MODIS" and delete the second sentence.
**Reply:** Updated.

132-135: Combine these two sentences.
**Reply:** This two sentences are updated as "*The VT map in 2015 shows that needleleaf evergreen forest (NEF), woody savannas (WDS), and grassland (GL), and croplands are the four dominant vegetation cover types over the CRB (Figure S1), and we study the impacts of wildfire over the NEF, WDS, and GL VTs.* "

153: Replace "burnt" with "burn"
**Reply:** Updated.

153-156: Start a new sentence after "respectively".
**Reply:** This part is updated as "*MTBS employs different integers to indicate burn severity categories, with values ranging from 1 to 4 representing unburned, low, moderate, and high severity, respectively. Consequently, the upscaling processes using the area-average remapping method produce floating-point numbers.*"

158: Delete "in 2015".
**Reply:** "in 2015" is deleted.

168: Replace "water" with "vapor".
**Reply:** Updated.

171: Replace "data sets" with "datasets".
**Reply:** Updated.

192-194: Strange wording and see reviewer 1's comment.
**Reply:** This part is clarified by adding descriptions in the end of Section 2.2.

Table 1: Agree with reviewer 1: move to SI. Also it's VPD, not WPD.
**Reply:** Table 1 is moved to SI with the data usage time frame and "VPD" updated.

209: Delete the extra period.
**Reply:** Deleted.

230: Capitalize Python.
**Reply:** Updated.

252-254: This should be in the methods section.
**Reply:** This part was updated by following the suggestions of both Reviewer #1 and Reviewer #2.

Figure 1: Agree with reviewer 1: better in the SI.
**Reply:** Figure 1 is moved to SI.

310-312: Strange wording.
**Reply:** This sentence is updated as "*In NEF, precipitation and VPD have importance scores of 0.3 for LAI resilience, while that of burn severity is 0.4 (Figure 4a).*"

316-317: First half of the sentence is poorly worded.
**Reply:** This sentence is updated as "*The importance scores for GPP and ET represented resilience show variations. However, the overall conclusion regarding the contributions of these three metrics to resilience values remains consistent across the VTs.*"

356: Replace "till 2020" with "within the post-fire study period".
**Reply:** Updated.

364: Delete the "y" on "fully".
**Reply:** Updated.

376: Delete "the result reveals".
**Reply:** "the result reveals that" is deleted.

378: Replace "the findings" with "previous work showing".
**Reply:** The first half of this sentence is updated as "*The results are consistent with the previous studies showing that forests tend to increase stomatal conductance and hydraulic efficiency, ...*"

395: Delete this first sentence.
**Reply:** To update the fifth paragraph of Introduction, we move the discussion of a previous study in Section 4.3. This sentence is updated correspondingly. Please check the first three sentences in Section 4.3 for the details.

396: "Though, burn severity is less important to grassland resilience,"
**Reply:** Updated

399: Use "interaction".
**Reply:** Updated.

407: Delete "across".
**Reply:** Deleted.

438-439: How does this study imply that these ecosystems may have extremely low chances of full recovery? 4 years is not a very long post-year period.
**Reply:** This sentence is updated as "*This study implies that with these changes, some ecosystems may need a long time frame to achieve full recovery. This prolonged recovery could keep carbon stocks at relatively low levels for decades to a century (Turner et al., 2019) and affect the ecosystem function and ecosystem–atmosphere interactions (Harris et al., 2016).*"

---

## Author Response (AR2)

**Reviewer # 2**

The authors have generally responded thoroughly to the first round of reviews. However, there is no discussion concerning the rates of increasing post-fire ET, GPP, and LAI in figure 1, and increasing resilience in figure 3 across the three VTs under high burn severity. While full recovery of a forest or savanna is not expected 5 years following a severe burn, it is interesting that GPP, LAI, and ET seem to increase at seemingly similar rates across pre-fire VTs despite different "importance scores" presented in figure 4.

**Reply:** This is a good point raised by the reviewer. We added the following discussion in L494_501:

*"Here, Figures 1 and 3 show similar recovery trends across NEF, WDS, and GL, while Figure 4 implies various importance scores of different factors in determining water dynamics and burn severity. We notice the differences in precipitation and temperature among these three VTs (Figure S4), with relatively drier and hotter conditions in GL, which underscores the role of climate in shaping the distribution of different VTs (Mather et al., 1968). In other words, the importance scores highlight the roles of different factors in recovery across VTs, which adapt to specific environmental conditions (Figures S1, S3, and S4) and exhibit their respective recovery rates."*

The added discussion concerning changes in pre to post fire VT transitions is appreciated, although figure 5 does not show where changes to VT have occurred. Another figure could be produced showing where VT transitions have taken place, however it is hard seeing pixel-scale details at the regional-scale of figure 5. Perhaps larger versions of these maps could be included in the supplemental materials.

**Reply:** Figure S12 is added in the SI with a smaller area and increased size of the panels. We will also upload the original high-quality (e.g., png, pdf) figures to the journal.

This version needs another round of proofreading. Some specifics are included below.
**Reply:** We did another round of proofreading, and the file with track changes is uploaded.

426 & 435 – These references to equations 1 and 2 are confusing.
**Reply:** The references for equations (1) and (2) are updated in these lines.

578-590: "One possible scenario is that in the next a few decades, NEF will regenerate, and the dominance of grass will decrease with the reestablishment of trees, which will compete with grassland for light and nutrients."
**Reply:** Updated.

Figure 5: The caption needs some attention.
- Delete "and with VT changes"
- Cropland is not in the legend, and does not appear to be in figure b.
**Reply:** The caption is updated.

605: component
**Reply:** "components" is updated to "component".

611: available ◊ active

**Reply:** "active" is used instead.

620-629: Sentence needs some reworking.

**Reply:** This part is updated as "*The similar responses of GPP and ET to fires could be associated with the tightly coupling between these two fluxes, which is governed by stomatal conductance. Stomatal conductance regulates both photosynthesis and transpiration (Knaue et al., 2020; Stoy et al., 2019), and the correlations between GPP and ET (Running et al., 2004). Partitioning the contributions of GPP and ET coupling, as well as the methods used to derive GPP and ET data, is beyond the scope of this study. Here, our third hypothesis is supported, and it suggests that resistance and resilience post-disturbance is highest for GPP and ET, and lowest for LAI across all VTs.*"